# Measuring Effects on Pain and Quality of Life after Abobotulinum Toxin A Injections in Children with Cerebral Palsy

**DOI:** 10.3390/toxins14010043

**Published:** 2022-01-05

**Authors:** Christian Wong, Ian Westphall, Josephine Sandahl Michelsen

**Affiliations:** 1Department of Orthopaedic Surgery, Copenhagen University Hospital, 2650 Hvidovre, Denmark; josephine.sandahl.michelsen@regionh.dk; 2Department of Paediatrics, Copenhagen University Hospital, 2650 Hvidovre, Denmark; ian.theodor.westphall@regionh.dk

**Keywords:** pain, musculoskeletal pain, cerebral palsy, children, Botulinum toxins type A

## Abstract

Sixty-seven percent of children with cerebral palsy (CCP) experience pain. Pain is closely interrelated to diminished quality of life. Despite this, pain is an overlooked and undertreated clinical problem. The objective of this study was to examine the analgesic effect of a single lower extremity intramuscular injection of Abobotulinum toxin A/Dysport in CCP. Twenty-five CCP with at least moderate pain (r-FLACC ≥ 4) during passive range of motion were included. Localized pain and pain in everyday living were measured by r-FLACC and the Paediatric Pain Profile (PPP), respectively. Functional improvements were evaluated by the goal attainment scale (SMART GAS). Quality of life was evaluated by either the CPCHILD or the CP-QOL. The subjects were evaluated at baseline before injection, then after 4, 12, and 28 weeks. Twenty-two subjects had a significant mean and maximum localized pain reduction (*p* < 0.001) at four weeks post-treatment in 96% (21/22). The reduction was maintained at 12 (19/19) and 28 weeks (12/15). Daily pain evaluated by the PPP was significantly reduced and functional SMART GAS goals were significantly achieved from 4 to 28 weeks. Quality of life improved significantly at four weeks (CPCHILD). Significant functional gains and localized and daily pain reduction were seen from 4 to 28 weeks.

## 1. Introduction

Cerebral palsy (CP) is the most common inborn neurological disease in children with a prevalence of 2.4 per 1000 live births [1]. CP is characterized as a movement disorder with muscle spasticity as the apparent sign, but CP also entails sensory deficits, as well as musculoskeletal and neuropathic pain [2,3]. The prevalence of pain is reported by caregivers as high as 67% [4,5], and the incidence increases with age and severity of the disease [6]. Pain can be as disabling as the movement disorder in itself [7,8], and self-reported pain is the primary determinator for diminished quality of life [9]. Despite the high prevalence of pain in CP, it is unrecognized and undertreated; thus, diagnostic strategies and treatment are inadequate, and the consequences are detrimental since chronic pain is associated not only with decreased physical functioning but also with disturbances in sleep, increased fatigue, and depression [4,5,10].

In our opinion, children with CP should be treated for their pain when treatments of evidence-based medical and non-pharmacological interventions are available [11,12]. However, evidence-based research in treatment is still warranted since pain in children with CP is undertreated; this is important since pain is closely interrelated to their wellbeing [4,5,11,13]. Botulinum toxins injections (BoNT) have a documented antispastic effect and have been utilized to modulate spasticity for two decades [14,15]. This subsequently improves functional mobility and ease of care in children with spasticity, but the analgesic effect of BoNT has been sparsely investigated with a moderate level of evidence for pain (OCEBM/Oxford Centre for Evidence-Based Medicine level II) [13,16]. For the adult population, BoNT has a documented relieving effect on spasticity-related pain in stroke, dystonia, and multiple sclerosis [17] (pp. 153–166). For the paediatric population, the analgesic effect for children with CP has been examined more sparsely. The pain-relieving effect of BoNT stems mostly from open studies and is examined as a secondary endpoint [8,11,13,15,17,18,19,20,21,22,23,24,25]. Four prospective randomized studies have evaluated the effect of BoNT on pain in children [8,22,24,25], and three open studies have also shown an analgesic effect [18,19,23]. The effect has been evaluated in relation to orthopaedic surgery or occupational therapy, short-term (three months), using botulinumtoxinA (OnabotulinumtoxinA in five studies, Aboa in one study, OnabotulinumtoxinA and AboA in two studies, and unspecified in two studies) and evaluated only by a few dimensions of pain [8,16,18,19,20,21,23]. Especially when examining pain in daily living, thorough and adequate pain evaluation in children with CP is challenging since they, besides having a physical disability, also are challenged with having cognitive, perceptive, and communicative impairments. This warrants appropriate pain evaluation methods to meet these challenges—preferably by self-reporting and using multiple and validated ‘pain assessment tools’ [5].

In this study, we set out to examine the potential medium-term (seven months) analgesic effect on everyday living of a single intramuscular injection of Abobotulinum toxin A/Dysport^®^ (AboA) using multifaceted pain assessment tools, evaluating for functional improvements, and changes in quality of life.

## 2. Materials and Methods

### 2.1. Population

The subjects were paediatric patients with predominantly spastic CP and belonged to all gross motor function classification system (GMFCS) levels. They were recruited from our hospital service area as a convenience sample. Inclusion criteria were children between two and eighteen years of age with spastic cerebral palsy who were botulinum toxin naïve or had a latency period of six months from their last injection and had at least moderate pain (≥4) during passive range of motion (pROM) when evaluated by the revised Face, Legs Activity, Cry, Consolability scale (r-FLACC). A medical doctor specializing in paediatric CP evaluated the patients for eligibility using clinical evaluations and medical charts before including them as subjects. Exclusion criteria were fixed contracture or severe athetoid/dystonic afflictions, or if modifications in ongoing treatment would affect pain status that had occurred three months prior to the AboA injection and during the project period (anti-spastic or pain medication given orally or by injection or if subjects had interfering surgical procedures performed).

The Regional Committee on Health Research Ethics and the National medical agency approved the study (H–17041772 and EudraCT number 2017-004497-33). We obtained oral and written consent from the caregivers, and the study was conducted according to national guidelines and the Helsinki Declaration. The study was investigator-driven and supported by Ipsen AB.

### 2.2. AboA Injections

A single ultrasound-guided intramuscular injection of AboA was administered without or under general anaesthesia by the discretion of the treating physician. Dosing was determined by the treating physician using 30 units per kilo (U/kg) and 15 U/kg for bilateral and unilateral CP, respectively, with a maximum dose of 1000 units. Small and large muscles were injected within a range of 3–6 U/kg and 8–12 U/kg, respectively. Small muscles were defined as an ultrasound-measured muscle thickness of a ‘diameter’ of less than 0.95 cm at the injection site and large muscles were defined as having a ‘diameter’ larger than 0.95 cm at the injection site [26,27]. Five hundred units of AboA were diluted in 2.5 mL of sterile NaCl in sterile syringes. Targeted muscles were determined after prior evaluation of pain response measured by r-FLACC during pROM.

### 2.3. Assessments

The subjects were assessed with observational pain, questionnaires pertaining to pain, function, and quality of life at baseline before injection and after 4, 12, and 28 weeks. Our primary endpoint was the change in pain status from baseline to the initial follow up at four weeks since AboA is considered to have an optimal effect at this time [14]. The subjects were monitored continuously for adverse effects as well as changes in medication or therapeutic interventions.

### 2.4. Pain Tools

The observational pain tools of the Paediatric pain profile (PPP) and r-FLACC were utilized to captivate different aspects of localized pain. A systematic pain interview was carried out by a single rater.

#### 2.4.1. Localized Muscular Pain Using the Revised Face, Legs Activity, Cry, Consolability Scale

Initial clinical evaluation entailed pROM of all muscles of the lower extremity to identify potential localized muscular pain using the r-FLACC. The r-FLACC scale is a validated behavioural pain intensity tool with five categories with a three-point ordinal scale (0–2), thus ranging from 0 to 10 possible points. Each category entails a description of behavioural signs in the facial expression, legs, activity, cry and consolability [28]. The r-FLACC scores were evaluated during the examination and were videotaped systematically using two iPads, thus enabling us to re-evaluate the subject in the frontal and sagittal view [29]. The caregivers added a unique descriptive ‘pain’ behaviour of the child to ensure that our ratings were individual and accurate. The localized pain was evaluated for the treated muscles during the follow ups. Since injections of AboA are presumed to have a localized effect, our primary endpoint was the change in the r-FLACC score. This was evaluated during passive range of motion of the targeted muscles and when AboA would peak in effect for spasticity reduction, namely at four weeks [14].

#### 2.4.2. Daily Pain Using the Paediatric Pain Profile

The PPP is a validated 20 item behaviour-rating scaled questionnaire for assessing pain behaviour and monitoring responses to treatment in children with neurological impairments [30]. Each item has a four-point ordinal scale (0–3) with a total score ranging from 0 to 60. The PPP is a caregiver-held tool to evaluate everyday pain.

#### 2.4.3. Pain Interviews

We conducted a structured interview with subjects and caregivers by asking them specific, relevant questions. We inquired specifically if the pain had impediments to the child’s activities in everyday living. Caregivers graded the impact of pain as if this prevented ‘no activities of daily living, few activities of daily living, some activities of daily living or most activities of daily living’. The caregivers were asked about the pain history after injection, i.e., when the pain subsided and then it returned as well as to the overall satisfaction with the treatment and if they warranted another treatment. Changes in medical treatment and rehabilitation during follow-ups were recorded.

#### 2.4.4. Functional Changes Evaluated by the Goal Attainment Scale (GAS) Using the Specific, Measurable, Achievable, Relevant, and Timed (SMART) Principles

During the initial interview, individual and relevant GAS goals using the SMART principles were set for each child. The goals were scaled from the status at baseline and before treatment (−1 or −2), the expected outcome (0), the favourable outcome (+1) and the most favourable outcome (+2). In this study, we set up two to three pain-related unweighted goals [31], evaluated them at each follow-up, and finally calculated the T scores [32].

#### 2.4.5. Quality of Life Evaluated by the Caregiver Priorities and Child Health Index of Life with Disabilities (CPCHILD) and Cerebral Palsy Quality of Life Questionnaire (CPQOL)

Quality of life was evaluated by validated questionnaires: CPQOL for GFMCS 1–2 and CPCHILD for GFMCS 3–5 [33,34,35]. The CPQOL has four patient appropriate versions (self-reported and proxy-reported for teenagers and children respectively), with seven domains each. In children, these were social wellbeing and acceptance, feelings about functioning, participation and physical health, emotional wellbeing and self-esteem, access to services, pain and impact of disability, and family health [33]. For teenagers, these were general wellbeing and participation, feelings about functioning, communication and physical health, school wellbeing, access to services, social wellbeing, and family health. These are rated on an ordinal scale and transformed to a standardized score. These are reported for each domain [34]. The original CPQOL was translated according to CPQOL translation guidelines into our native language (Danish) for this study.

The CPCHILD questionnaire consists of six domains with a total of thirty-six items [35]. The domains are pertaining to personal care (8), positioning, transfer, and mobility (8), communication and social interaction (7), comfort, emotions, and behaviour (9), health (3), and overall quality of life (1). These are rated on an ordinal scale and as standardized scores. Each domain is reported separately and as a total score (0–100). Some items also have a modifier that grades the level of assistance from the caregiver.

#### 2.4.6. Statistical Analysis

The sample size was estimated to be sixteen subjects. This was determined using Gpower© with a power (1-β) of 0.95, an α level of 0.05, a change in pain level by the r-FLACC in a two-tailed match pair *t*-test and based on the standard deviations of 2.7 for video evaluation from Malviya et al. (2006) [28]. The estimated dropout at four weeks was 7 subjects; thus, at least 24 patients had to be included. The overall comparisons were between the data at baseline and follow-ups. Shapiro–Wilk for normality was utilized to determine normal distribution. Data from r-FLACC, PPP, and CPCHILD scores were continuous variables and analysed using the paired *t*-test. Ordinal variables such as impact on activity and SMART goals were not normally distributed and analysed using the Wilcoxon signed-rank test. *p*-values of ≤0.05 were considered statistically significant. Bonferroni corrections were applied to the r-FLACC since we analysed the mean and maximum scores (*p* < 0.05/2 = 0.025) and to CPCHILD when analysing the six domains (*p* < 0.05/6 = 0.008) to reduce type I error. All tests were performed using IBM SPSS Statistics, Version 25 (IBM, Richmond, VA, USA).

## 3. Results

### 3.1. Subjects

Children with CP in our general service area were screened for eligibility. Fifty-one of them had pain and were contacted through their caregivers for inclusion after a formal invitation by letter. After caregivers accepted participation, their medical records were screened according to the inclusion and exclusion criteria. If still eligible, the patients were examined for lower extremity muscular pain by the r-FLACC during pROM (≥4). Twenty patients had pain levels too low to participate, four patients declined to participate and two had other ongoing changes in medical treatment, hence were excluded. Twenty-five patients with spastic cerebral palsy and an r-FLACC pain level larger than four were included as subjects and received a single injection of AboA. The majority had previous BoNT–A treatment (21/25). Twenty-two subjects were evaluated at a 4-week follow up, 20 at 12 weeks, and 15 at 28 weeks. The total withdrawals were nine, whereas seven had a good but subsiding pain-relieving effect and there was no effect on pain in two subjects. One subject was treated with AboA with good effect, but the storage temperature of the AboA was not monitored adequately before injection, and thus is excluded. Figure 1 shows a flow chart of the history of the subject’s participation and exclusion.

The average age was 9.1 years with a range from 2 to 17 (SD: 3.6) for the 22 subjects that were still included at the 4-week follow up. The gender ratio (female:male) was 8:14. They were classified as unilateral:bilateral with a ratio of 3:19 (hemilegia:3, diplegia:9 and tetraplegia:10). Figure 2 shows a characterisation of the subjects according to the GFMCS and other classifications.

The ultrasound-guided injections with AboA were carried out by a single doctor at the primary centre except for three injections, which were performed by two neuro-paediatricians. All procedures but one were performed under general anaesthesia. The average injected dose was 21.8 units/kg of AboA. This was 15 units/kg and 25.3 units/kg for unilateral and bilateral CP, respectively. The AboA injections were injected into a combination of one muscle (Adductor Femoris) for two subjects, two muscles groups (often Psoas of the hip, Adductor Femoris or the Gastro and Soleus muscle of the Gastrosoleus muscle of the calf) in seventeen, and three muscle groups (Psoas the hip, Adductor Femoris combined with either Gastrosoleus of the calf or medial hamstrings or Rectus Femoris) in three. The injected muscles are shown in Figure 3. After the injections, the subjects had additional physiotherapy combined with their ongoing rehabilitation. The medical status in general and at baseline are shown in Table 1. However, the latter part of this study was conducted when the COVID-19 pandemic demanded the closure of non-vital clinical contacts. Consequently, we conducted four video/telephone consultations instead of in-person follow-up visits, five follow-ups were postponed (median delay four weeks, ranging from three to eight weeks) and some of the planned post-injection physiotherapies were cancelled during this period.

### 3.2. PAIN Assessment and Questionnaires

#### 3.2.1. r-FLACC for Localized Pain

We found a pain-relieving effect on localized muscle pain in 96% (21/22) of the subjects at four weeks with a significant mean and maximum r-FLACC score reduction of 2.53 points (SD:2.07, *p* < 0.001) and 2.57 points (SD:2.42, *p* < 0.001), respectively. The significant pain reduction was maintained in the period 12 (19/19, missing data in 1; mean 3.02 (SD:1.91), *p* < 0.001; max 2.25 (SD:2.60), *p* = 0.001) and 28 weeks (12/15; mean 2.33 (SD:2.50), *p* = 0.006; max 2.46 (SD:2.33), *p* = 0.002). The changes in pain level evaluated by three pain scales are shown in Figure 4.

#### 3.2.2. Paediatric Pain Profile for Daily Pain

The PPP was significantly reduced at four weeks with a mean reduction of 8.77 points (SD:12.11, *p* = 0.003). This reduction in PPP was maintained with 9.28 points (SD: 10.65, *p* = 0.002) and 7.13 (SD: 11.85, *p* = 0.03) at 12 and 28 weeks, respectively. The changes in PPP at baseline and after 4, 12, 24 weeks are shown in Figure 5.

#### 3.2.3. SMART GAS for Functional Changes

The pain-related SMART goals were related to improvements in sleep (40%), standing in a standing frame (10%), falling/tripping (6.7%), gait-related (3.3%), pain-related (16.7%), and other goals (16.7%). Table 2 shows an overview of the number and types of SMART goals. There was a significant increase in the mean T score, which changed from 33.25 at baseline to 52.31 at four weeks (Z = 3.61, *p* > 0.001). Seventy-three percent (16/22) achieved one or more goals to the expected or higher level. This was maintained significantly at 12 (T = 50.31, Z = 3.08, *p* =0.002) and 28 weeks (T = 46.85, Z = 2.535, *p* = 0.011). The changes in the distribution of achieved goals during the 28 weeks and changes in T score are shown in Figure 5 and Figure 6, respectively.

#### 3.2.4. CPQOL (GMFCS I-II) and CPCHILD (GMFCS III–V) for Quality of Life

Quality of life evaluated by the CPchild questionnaire improved significantly at four weeks. The increase in the total score changed from 59.53 ± 6.94 to 63.12 ± 7.90 (*p* < 0.001). The effects were not significantly sustained at 12 and 28 weeks when Bonferroni correction was applied (significance level after correction *p* = 0.016). No significant effect was seen in the seven subdomains of CPChild after the Bonferroni correction. The changes in quality of life by the CPChild before injection at baseline and after 4,12, and 28 weeks are shown in Figure 5.

The response rate for answering the QOL questionnaires at follow-ups ranged from 84% to 92% for CPCHILD, and 6 out of 8 responded to the CPQOL (4 proxy-reported children and 2 self-reporting teens). For CPQOL, there were improvements in 2/7 domains for caregivers’ evaluation for children and 0/5 for self-evaluation for teenagers.

#### 3.2.5. Overall Satisfaction, Impact on Activities, Additional Treatment, and Adverse Effects

Eighty-six percent of caregivers (19/22) were overall satisfied with the treatment and the effect of the treatment and 86% (19/22) warranted and/or had treatment again. The caregiver reported that the effect on pain began to subside or end within the period of 12 and 28 weeks (median). Table 3 shows the reported change in caregivers’ perceived effect on pain level. The caregiver reported that the pain before treatment had no impact on activities of daily living for 14% (3/24) and 73% had an impact on some or most activities (16/24) within the last week. There was a significant improvement in impact on activities at four weeks (Z =2.610, *p* = 0.009). This significant effect was sustained at 12 weeks (Z = 2.547, *p* = 0.011) and 28 weeks (Z = 1.807, *p* = 0.07) but not significantly at 28 weeks.

Nine subjects had additional treatment and were excluded before the end of the study. All nine received another AboA injection prematurely to 28 weeks, 2 combined with either an increased dose of oral Baclofen and 2 combined with orthopaedic muscle lengthening for two subjects.

Eleven subjects experienced adverse events during the study. In total, there were sixteen adverse events. The majority were related (12). These were temporary mild (5) and moderate (1) muscle weakness, mild nausea (2), and mild soreness/local bruising (2) at the injection site. Two had moderate pain (2) at the injection site that subsided within the first week. An additional three caregivers reported unrelated adverse events with weight gain (1), mild dyspnoea (1), and mild diarrhoea (1) during the study. At the study initiation, one subject (1) had an uneventful admission overnight since the ultrasound-guided injection of AboA was delivered close to an intramuscular vessel. This was classified as a serious adverse event. There was otherwise no expected or unexpected serious adverse events or reactions.

## 4. Discussion

In this study, we found a significant and clinically meaningful pain-relieving effect after one injection treatment of Abobotulinum toxin A (Dysport^®^) for localized and daily pain. The measured effect was highest at 4 weeks, which was significantly maintained at 28 weeks. The effect on pain was sustained significantly during the study, but the clinical meaningful peaked at four weeks and was maintained at 12 weeks. The effect subsided between 12 and 28 weeks, hence several patients requested another treatment. It seemed ethically unviable to continue the observation period since the treatment had a major effect on the ability to participate, physical activity and general well-being and we felt obliged to re-treat them. The patients that completed the study had less pain at 28 weeks than initially, but pain re-emerged after 12 weeks in general. We had two late responders with effect when measured at 12 weeks and one had a lasting effect over one year. The pain reduction also mediated significant and clinically meaningful patient-relevant functional improvement as well as an impact on quality of life at four weeks.

This is the first study to evaluate the effect of a single injection of AboA on pain with several dimensions of pain, function, and quality of life. This study evaluated the pain over a longer period than previous studies, thus enabling us to establish a clinical meaningful pain-reducing effect at least after 12 weeks. Previous studies have demonstrated an effect for spasticity for a period of less than 23–28 weeks [36,37], thus we chose to evaluate for pain at 28 weeks to supersede this period and capture when the pain would subside.

Though the pain-relieving effect has been established in four prospective placebo randomized trials for children with CP [8,22,24,25], Tilton et al. (2017) demonstrated pain reduction as a secondary and surrogate endpoint using the goal attainment scale [24]. Barwood et al. (2000) found a 74% reduction in pain and pain medication for children in an operative setting [22]. Copeland et al. (2014) found a significant reduction of one-third in PPP four weeks after injection but was unable to detect a significant reduction in PPP when compared to a sham injection in a subgroup of patients with mild pain. In this study, pain was not the primary indication for treatment and was evaluated as a secondary endpoint [8]. We found a 44% and 32% reduction in r-FLACC and PPP, respectively. Rivard et al. (2009) used telephone interviews for pain evaluation and found that 62% of parent-proxies reported that their children did not experience pain one month after AboA injections when compared to the placebo [23]. Two previous studies have reported a reduction in PPP after a single injection of AboA of 34% and 17% at four and sixteen weeks [8] and 81% at 12 weeks [18]. We demonstrated a pain reduction in PPP with 32% and 25% at four and 12 weeks, respectively. Both studies also evaluated functional effects. Using the Canadian Occupational Performance Measure, Copeland et al. (2014) demonstrated faster and longer-lasting improvements in ease of care and comfort, tolerating splints, and hygiene [8]. Though these effects were not maintained at sixteen weeks. Lundy et al. (2009) used GAS and found patients improved in sleeping, seating, handling, and movement, which subsequently led to a perceived improvement in developmental achievements [18]. We concur with the above study with similar findings with significant functional improvements for sleep, mobility-related goals, and specific pain-related problems when evaluated by GAS. These functional improvements seem to be maintained when the pain-relieving effect subsided.

In our study, QOL improved significantly at four weeks. The effect on QOL was not maintained at 12 and 28 weeks as well as for the subdomains of CPChild after Bonferonni correction. Our results for CPQOL were diverse without clear improvements. Our findings concur with Copeland et al., who also only demonstrated significant improvement in CPCHILD at four weeks, but not at sixteen weeks in either CPQOL or CPCHILD [8]. Though they found children had improvements in specific aspects of their life, this was not reflected in a general, more holistic measure of wellbeing across broad domains. We observed that patient satisfaction was most evident in children, who expressed that their pain influenced their life negatively and when having higher pain intensities. After four weeks, caregivers spontaneously shared that they now realized how much their child’s pain had affected their life and saw those lost functions returned after approximately one week. These improvements were unexpected. This was reflected in our high overall satisfaction which led some caregivers to warrant another treatment. However, the pain-relieving effect subsided after 12 weeks, and we noticed that caregivers vigilantly pursued another treatment to a degree that we have not experienced when treating spasticity.

In conclusion, a significant localized and daily pain reduction was seen from baseline to 28 weeks after treatment with AboA. This facilitated functional gains at 12 weeks as well as the quality of life at 4 weeks after treatment. Our results are in concurrence with previous studies, but in general, our improvements in dimensions of pain were (slightly) less. In this study, AboA injections had a clinical meaningful pain-reducing effect at 12 weeks which then subsided. We agree with Copland et al. (2014) that therapy provides these gains, but even better results can be achieved by caregivers’ involvement or by adding other specific pain reduction therapies [8]. In our experience, the need for injection therapy reoccurs after 12 weeks when the clinical meaningful (but not statistical) effect subsides. In this study, we also noticed two late responders (2/22), where the effects occurred when evaluated at 12 weeks and where no other interventions could explain this. This discouraged the subjects and caregivers initially, but eventually, they reached similar levels of satisfaction as the other participants. Nine subjects reported no pain or less pain at the end of the study, but muscle tightness reoccurred and declined in the attained functional improvements.

Since previous studies already have demonstrated a pain-reducing effect of botulinumtoxinA, using a placebo control group could be construed as unethical. One might argue that the pain-relieving effects in this study could be ascribed to placebo, but the magnitude of pain reduction response and consistency of effects across almost all endpoints contradict this. Previously, it has already been construed that using a placebo design would be ethically unfeasible due to this apparent good and documented effect [23]. The choice of preparation of botulinumtoxinA was considered less important since Lundy et al. (2009) demonstrated them to be equipotent [18]. The pain reduction could, in principle, be ascribed to other pain-relieving medication. However, no such changes occurred at the four-week follow up, except for one patient, who discontinued the oral muscle relaxant (Baclofen) because of the study. Two patients increased oral Baclofen before the 12-week follow up, when the pain-relieving effect of AboA subsided, which led to their measurements being discarded henceforward. At inclusion, the subjects were already optimized in their pharmacological treatment, which was prescribed by their treating paediatric physician. We considered them refractory to the ‘best practice of pharmacological pain treatment’.

Using an observational pain evaluation as the r-FLACC might induce reporting bias, since we were unable to blind the evaluation as well as the risk of a type one error due to the numerous evaluated parameters. We utilized a systematic setup for this pain evaluation using video recording. This enabled us to perform valuable re-evaluations retrospectively as well as discover discrete signs of pain in facial micro-expressions that otherwise easily would be missed. One rater examined the subjects, and one rater always evaluated all videos for a specific patient to ensure consistency. We accustomed ourselves to describe our observations orally during recording, which was valuable when re-evaluating the videos. To reduce the risk of a type one error, we utilized a Bonferroni correction in our statistical evaluation when appropriate. This diminished our significance level for the domains of CP child to less than 0.008. Moreover, the COVID-19 pandemic inflicted impediments in some of our evaluations with postponements and we had to evaluate some of the subjects using other methods, i.e., caregivers were asked to conduct the examinations. This might have affected our findings, but not for the primary outcomes at four weeks. The evaluations afterwards were influenced by these impediments, and for example, in one of the postponed follow-ups, the caregiver reported a pain-relieving effect at 12 weeks, but this subsided by the postponed examination at sixteen weeks (reporting bias).

Due to the heterogeneous nature of children with CP, a multitude of assessment tools are required to embrace the specific deficits of a child (i.e., inability to self-report, using observational tools for non-communicative children, or sometimes using combination tools) [4,5]. Preferably, these tools should be validated. Since most of our participants had cognitive and communication deficits, several observational pain tools were used to measure both localized pain and daily pain. We chose the validated observational tools of r-FLACC for localized pain and PPP for daily pain. Using these enabled us to compare our findings to previous studies. Other appropriate and validated tools, such as the NCCPC-R, PPIS, and PPQ exist and could have been chosen, thus examining different dimensions of pain [5]. Moreover, we prioritized that only one rater evaluated the subjects and focused on changes in pain level over time since a previous study has shown poor agreement between raters as well as between raters and self-estimation of pain [25].

Our study could be influenced by selection bias due to our exclusions and dropouts. We had three initial dropouts. Two patients did not experience the expected effects and did not attend the follow-up visits. One patient had a good response and a pain-relieving effect but was excluded since there was no documentation for the right storage for that package of study medicine. We also excluded seven patients due to ‘early’ AboA pain treatment. This probably has affected our findings, maintaining a more positive pain-relieving effect with relatively minor decreases in r-FLACC and PPP at 28 weeks. When using the imputation method of “Last Observation Carried Forward” to compensate for the missing follow-up data, the significant effect on localized pain was maintained. Thus, we maintain our conclusion that AboA has a clinically meaningful effect at four weeks. Specifically for the two subjects, who dropped out with no apparent effect, this could have been caused to the heterogeneity of causality of their pain. Retrospectively, we found that some of the subject’s perceived pain was caused by a combination of both localized muscular and neuropathic pain as described by Blankenbrug et al. (2018) [3]. This might also have been attributed to potential gastrointestinal-related pain [4]. Even after the reduction of localized muscular pain, the neuropathic pain or gastrointestinal-related pain would remain. In this study, this was reflected in their overall lower level of treatment satisfaction for these subjects. As a clinical observation, some of our subjects and caregivers expressed a relatively higher pain level than reflected in our evaluations of localized pain. We interpreted this as their pain had a component of central mediated neuropathic pain, where our local intramuscular injection therapy would have less effect [38,39].

In this study, we utilized the SMART goals measured by GAS to describe the functional changes but did not quantify the achieved goals formally. However, the functional improvements were confirmed in the structured interviews, where especially having improved sleep, having less pain when using orthosis or shoes, and showing improvement in standing and walking function were seen. Tilton et al. (2017) demonstrated similar functional improvements using GAS in a placebo-controlled design [23], thus affirming our findings. Copeland et al. (2014) reported that such improvements had an impact on the whole family’s quality of life [8]. In these interviews (and in our study), caregivers also described that lost functions, which had disappeared slowly, suddenly re-emerged when the subjects were unaffected by pain. We found that pain seems to have an impact on activities of daily living.

## 5. Conclusions

In conclusion, a significant and clinically meaningful, pain-relieving effect of one injection treatment of Abobotulinum toxin A (Dysport^®^) for localized and daily pain had a significant effect for 28 weeks with the highest measured effect at 4 weeks. This mediated improved function and overall satisfaction and had a short-term effect on quality of life. The clinically relevant, pain-relieving effect subsided after 12 weeks to a level warranting another treatment.

## Figures and Tables

**Figure 1 toxins-14-00043-f001:**
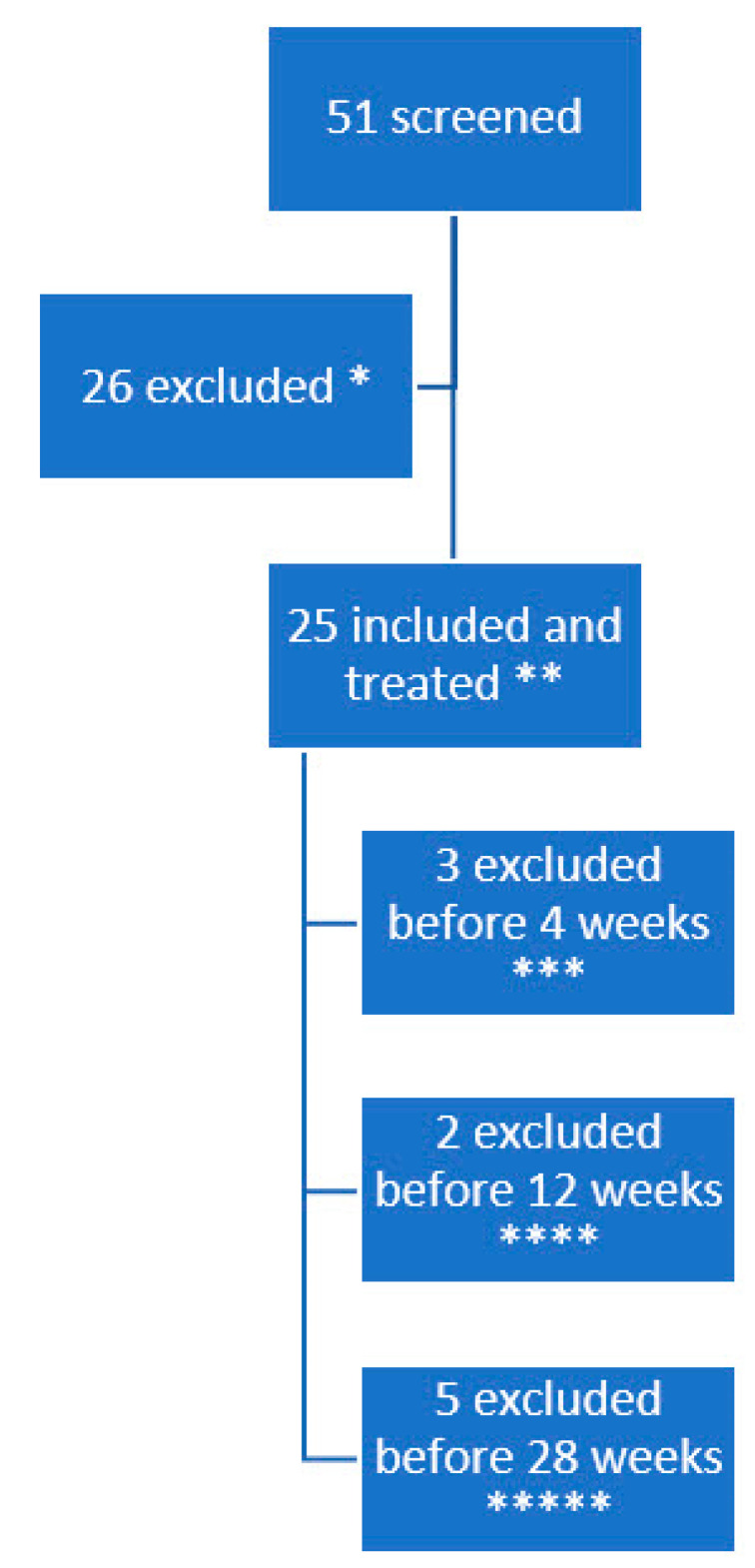
Timeline of the history of the subject’s participation. * 20 subjects due to r-FLACC < 4, 2 due to other treatments, 4 did not want to participate, ** with r-FLACC > 4, *** Two due to no effect and treated shortly after with another botulinum toxin injection treatment with higher dose and other muscles. One had a good effect but was treated with AboA that was not monitored adequately and was thus excluded. **** Two excluded since pain reoccurred and had additional pain treatment and ***** Five subjects were excluded due to deteriorating effect and were excluded to have another AboA injection.

**Figure 2 toxins-14-00043-f002:**
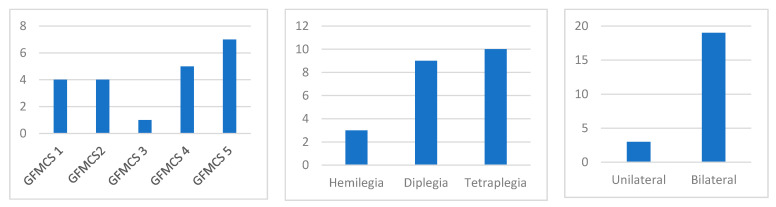
Characterisation of the subjects according to the GFMCS and other classifications of the subjects that were still included at the four-week follow up.

**Figure 3 toxins-14-00043-f003:**
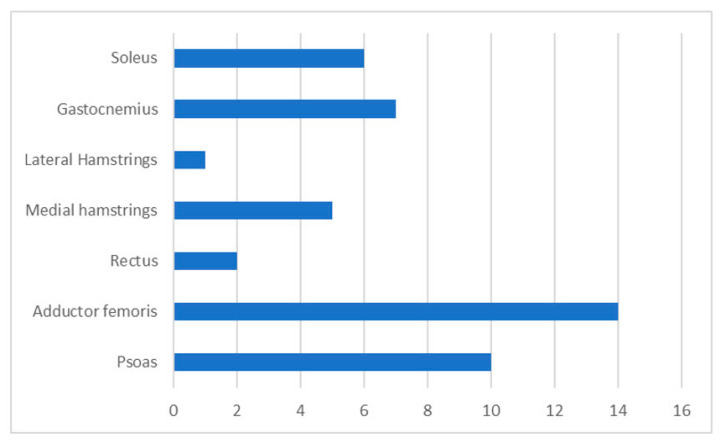
Distribution of the injected muscles of the lower extremity.

**Figure 4 toxins-14-00043-f004:**
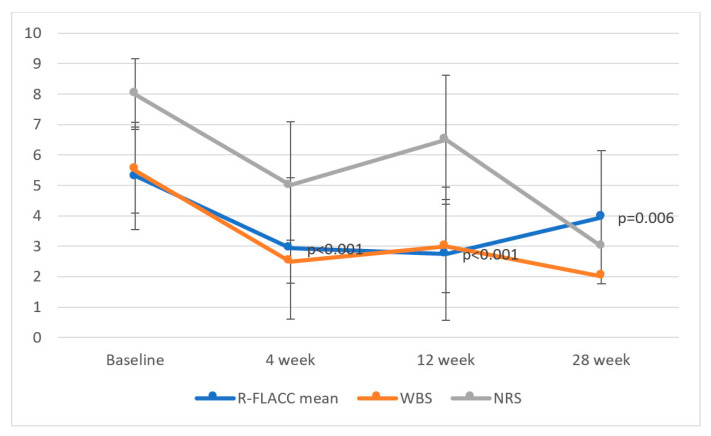
Observed and reported pain by mean r-FLACC, Wong–Baker, and the numeric pain scale before injection at baseline and after 4, 12, and 28 weeks (error bar = 1 standard deviation).

**Figure 5 toxins-14-00043-f005:**
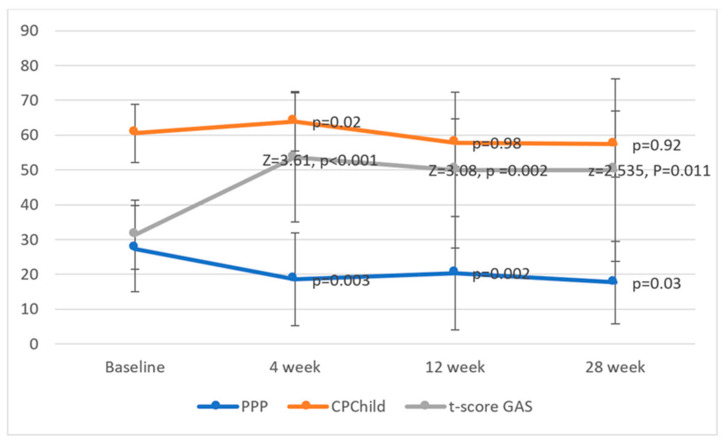
Reported pain by paediatric pain profile (PPP), T score for GAS and quality of life by the CPCHILD before injection at baseline and after 4, 12, and 28 weeks (error bar = 1 standard deviation for CPCHILD and PPP, 1 IQR for the T score).

**Figure 6 toxins-14-00043-f006:**
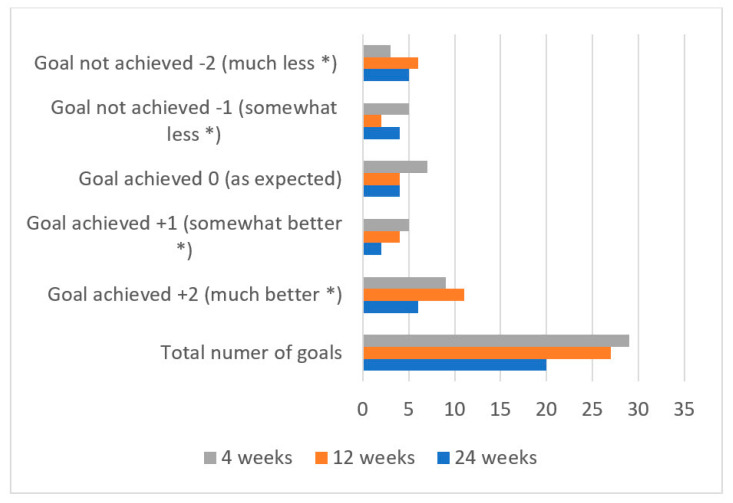
The distribution of achieved goals during the 28 weeks after injection. A GAS value of 0, +1, and +2 was considered as a goal achieved and a GAS value of −1 and −2 were not achieved. * Than expected.

**Table 1 toxins-14-00043-t001:** Status of medication at baseline, ongoing physiotherapy, and previous surgeries. * Paracetamol excluded. ** How many previous botulinum toxin injections on average. *** Region of surgery.

Overview of Medical and Surgical Tretaments	%
Receive other medical treatment	73
Muscle relaxant	32
Sleep medication	14
Medication for epilepsy	41
Pain medication *	27
Paracetamol	41
Obstipation	32
Pervious Botulinum toxin	82
How often **	5
Previous surgery	63
hip ***	36
thigh ***	5
knee ***	32
calf ***	32
Physiotherapy	96
How often **	3

**Table 2 toxins-14-00043-t002:** Overview of the SMART goals; types of goals according to ICF classification and evaluated by the goal attainment scale. * Quality of sleep, onset of sleep, awakenings, amount of sleep ** in a standing frame, *** frequency of falls, **** diaper change, ***** better ROM, putting on shoes, good/bad days and ****** painful days, pain when performing sports, pain when placed in a resting position, pain putting on orthosis, and pain during the examination.

ICF Code	Description	Frequency
b134	Sleep function *	12
d4154	Maintaining a standing position **	3
d498	Mobility, other specified ***	2
b	Body function ****	1
b770	Gait pattern functions	1
	other *****	3
b280	Sensation of pain ******	7
Total		29

**Table 3 toxins-14-00043-t003:** Caregivers’ perceived effect in pain level after treatment. * two excluded due to no effect, ** seven excluded due to additional treatment with AboA, tenotomy surgery, or oral Baclofen.

	Residual Effect
No initial effect	2/24
At 4 weeks	17/22 *
At 12 weeks	13/20
At 28 weeks	9/15 **

## Data Availability

For further interest, inquiries for data can be sent to the corresponding author.

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
