# Peer review of "Measuring Effects on Pain and Quality of Life after Abobotulinum Toxin A Injections in Children with Cerebral Palsy"

_toxins, 2022, doi:10.3390/toxins14010043_

Round 1
Reviewer 1 Report
The authors investigated the effects on pain and QOL after the Abobotulinum toxin A injection in children with cerebral palsy (CP). The main conclusion was that a significant pain reduction was seen from baseline to 28 weeks after injection of a single dose of AboA. The pain-relieving effect began to subside after the 12th week. The pain-relieving effect facilitated the functional gains during the study period and improve QOL four weeks after treatment. I have the following comments and suggestions.
- Abstract: According to the instruction for authors, the abstract should be a total of about 200 words maximum. Obviously, the word number in the abstract was over 200 words.
- Introduction: Good review about epidemiology, pathophysiology and previous treatment of CP.
- Introduction: OnabotulinumtoxinA (BOTOX) is more commonly used than AboA. Why used AboA, but not OnabotulinumtoxinA?
- Why the effective duration of AboA is about 7 months? Reference is needed.
- Inclusion criteria: When these patients were evaluated by r-FLACC, did they take any medication or non-pharmacological treatment? If yes, how to evaluate they were refractory to conventional treatment?
- Why choose 2 years old patients? They seemed too little to be evaluated.
- AboA injection: Why general anesthesia was needed for just a single injection?
- What are small muscles and what are large muscles? Definition is needed.
- When using r-FLACC, if the score was different from children and care-giver, which score was used?
Author Response
Thank you for an excellent summary and most relevant comment to the article ' Measuring effects on pain and quality of life after Abobotulinum Toxin A injections in children with cerebral palsy.'
We have revised the manuscript following your comments and will describe this in detail:
- Abstract: According to the instruction for authors, the abstract should be a total of about 200 words maximum. Obviously, the word number in the abstract was over 200 words.
Good point. The abstract is revised and shortened to 200 words as suggested to ensure and improve its readability.
- Introduction: Good review about epidemiology, pathophysiology and previous treatment of CP.
Thank you for this kind comment.
- Introduction: OnabotulinumtoxinA (BOTOX) is more commonly used than AboA. Why used AboA, but not OnabotulinumtoxinA?
Excellent and important point. OnabotulinumtoxinA is widely used and well known for spasticity treatment as well as for pain. In the nine studies examining the pain related to hypertonia for children with CP, two studies did not specify the sub-type of BTX or brand, five examined OnabotulinumtoxinA exclusively, one study examined AbobotulinumtoxinA, two examined both. For this reason, we set out to examine the effects on pain for AbobotulinumtoxinA. We have added two clarifications to the manuscript and the latest review from 2021 in the article, namely:
This subsequent improves functional mobility and ease of care in children with spasticity, but the analgesic effect of BoNT has been sparsely investigated with a moderate level of evidence for pain (OCEBM/Oxford Centre for Evidence-Based Medicine level II)
(l.52)
The effect has been evaluated in relation to orthopaedic surgery or occupational therapy, short-termed (three months), using botulinumtoxinA (OnabotulinumtoxinA in 5 studies, Aboa in 1 study, OnabotulinumtoxinA and AboA in 2 studies, and unspecified in 2 studies) and evaluated only by a few dimensions of pain
(l. 60-61)
and the added reference:
Almina S, Karile Y,Audrone P,Indre B. Analgesic effect of botulinum toxin in children with cerebral palsy: A systematic review. Toxicon. 2021 Aug;199:60-67.
The previous study of Lundy et al. (2009) has demonstrated a significant reduction in pain for both OnabotulinumtoxinA and AbobotulinumtoxinA. The reduction in pain score with an injection of BTX did not depend on the preparation used, and interestingly neither did it correlate with the dose of either preparation.
We have added this to the discussion:
Since previous studies already have demonstrated a pain-reducing effect of botulinumtoxinA, using a placebo control group could be construed as unethical….The choice of preparation of botulinumtoxinA was considered less important since Lundy et al. (2009) demonstrated them to be equipotent [18].
(l. 405-206 and l. 411-412)
Lundy CT, Doherty GM, Fairhurst CB. Botulinum toxin type A injections can be an effective treatment for pain in children with hip spasms and cerebral palsy. Dev Med Child Neurol. 2009 Sep;51(9):705-10. doi: 10.1111/j.1469-8749.2009.03315.x. Epub 2009 Apr 21. PMID: 19459910.
- Why the effective duration of AboA is about 7 months? Reference is needed.
Excellent point. We wanted to ensure that we monitored the full duration of potential pain relief.
Previously, Esquenazi et al. (2020) have shown an effect up to 28 weeks for lower limb spasticity in pediatric CP and Nestor et al (2011) showed a partial effect for ABoA of 160 days. We wanted to ensure that we superseded these periods.
We have clarified this in the discussion and added the references in the manuscript:
This study evaluated the pain over a longer period than previous studies, thus enabling us to establish a clinical meaningful pain-reducing effect at least after 12 weeks. Previous studies have demonstrated an effect for spasticity for a period of less than 23-28 weeks [36,37], thus we chose to evaluate for pain at 28 weeks to supersede this period and capture when the pain would subside.
(l. 348-505)
The references added:
Nestor MS, Ablon GR. Duration of action of abobotulinumtoxina and onabotulinumtoxina: a randomized, double-blind study using a contralateral frontalis model. J Clin Aesthet Dermatol. 2011 Sep;4(9):43-9.
Esquenazi A, Delgado MR, Hauser RA, et al. Duration of Symptom Relief Between Injections for AbobotulinumtoxinA (Dysport®) in Spastic Paresis and Cervical Dystonia: Comparison of Evidence From Clinical Studies. Front Neurol. 2020;11:576117. Published 2020 Sep 25. doi:10.3389/fneur.2020.576117
- Inclusion criteria: When these patients were evaluated by r-FLACC, did they take any medication or non-pharmacological treatment? If yes, how to evaluate they were refractory to conventional treatment?
Excellent point. The subjects were in medical treatment for pain as seen in the table below (table 1 from the article):
(see in word file)
The subjects had pain despite their prescribed pain medication and non-pharmacological treatment. Typically, they had been optimized before their enrollment in the study with various medical interventions for pain by their treating pediatric physician, and even might have been referred to this study by them, since the pain condition was resistant to the prescribed medicine and the experienced pain superseded the effect of prior medical treatment. We considered that they were refractory to the best practice of currently accepted medical treatment for pain.
In the discussion, we have added
At inclusion, the subjects were already optimized in their pharmacological treatment, which was prescribed by their treating pediatric physician. We considered them refractory to the ‘best practice of pharmacological pain treatment’.
(l. 417-420)
If we wanted to examine the effect of an AboA injection even more specifically, we should have discontinued the prescribed medicine. However, we found this ethically unfeasible, and would probably not have obtained ethical approval for such a change in intervention by our ethical board. Instead, we rigorously maintained (meaning: did not increase) the current medical treatment and evaluated the changes in pain level by the various pain scores over time.
- Why choose 2 years old patients? They seemed too little to be evaluated.
Good point. Usually, the observational pain evaluation tool ‘mod. r-FLACC’ has good to excellent reliability and validity for pain assessment in children with cognitive impairment, and is usually used for somewhat older children, but are validated as low as 22 months as examined by Crelloin et al. (2021). We also found it justifiable to use this observational tool for younger children since this is routinely used for pain evaluation even for infants in our pediatric department.
Crellin D, Harrison D, Santamaria N, Babl FE. Comparison of the Psychometric Properties of the FLACC Scale, the MBPS and the Observer Applied Visual Analogue Scale Used to Assess Procedural Pain. J Pain Res. 2021;14:881-892. Published 2021 Mar 31. doi:10.2147/JPR.S267839
- AboA injection: Why general anaesthesia was needed for just a single injection?
Thank you. We have not informed you fully about this matter, and we apologize; this was unintentional. In this study, the injections were performed without or under general anaesthesia at the discretion of the treating physician. However, all but one were performed under full anaesthesia, due to the treating physician being unable to sedate the child to cooperate with the anaesthesia procedure. We believe the need for anaesthesia is dependent on the experience of the treating injector, especially for adults and older children. The more experienced the injector is the less need for sedative procedures. In our study, the majority of the children were either very young or had cognitive impairments, thus did not understand the circumstances of the procedure and as a consequence became frightened and could not corporate with the procedure without anaesthesia. In these instances, general anaesthesia was purposeful, but as the reviewer rightly infer, that it has to be weighted towards the risks of complications and side effects.
We have added to the manuscript in the method and materials section:
A single ultrasound-guided intramuscular injection of AboA was administered without or under general anaesthesia by the discretion of the treating physician.
(l. 93-94)
We have added to the manuscript in the results section:
The ultrasound-guided injections with AboA were carried out by a single doctor at the primary centre except for three injections, which were performed by two neuropaediatricians. All procedures but one were performed under general anaesthesia
(l. 223)
- What are small muscles and what are large muscles? Definition is needed.
The reason to define muscle as either small or big was to prevent diffusion of AboA. Diffusion is influenced by several factors such as dose, concentration, volume, rate of injection, needle size, the distance of needle tip from the neuromuscular junction, number of injections, target muscle selection, the presence of muscular fascia, the presence of tissue damage at the injection site, muscle contraction following injection, and the protein composition and molecular size of the BoNT formulation. However, dose, concentration, and volume are probably the greatest contributors (Brodskt et al. (2012)).
In regards to the evaluation of the target muscle, this is based on clinical experience and experimental studies as described by Kinett et al (2004) and Sätilä (2020):
…directly detecting BTXA spread in the human muscle, based on the animal and human data available and clinical experience…
… As BtxA is injected into a muscle, there is an immediate diffusion of toxin in the muscle within a few centimetres of the needle tip...
This inferred to us, that small muscles should be defined by size, which practically could be evaluated by ultrasound. Small muscles were defined an ultrasound-measured muscle thickness of a ‘diameter of less 0.95 cm’ on ultrasound at the injection site. In children with CP under the age of eight years, the triceps surae muscles are surprisingly thin: medial GC 0.9 cm, lateral GC 0.75 cm, and soleus 0.98 cm, on average. In this study, the soleus muscle was considered large, whereas the psoas muscle was injected in the area of the anterior to the femoral head. Here the psoas muscle is much less than 0.9 cm. This was considered small. We are aware that the muscle thickness is a ‘pseudo’ measure of the volume of the.
We have added:
Small and large muscles were injected within a range of 3-6 U/kg and 8-12 U/kg, respectively. Small muscles were defined as an ultrasound-measured muscle thickness of a ‘diameter’ of less 0.95 cm at the injection site and large muscles were defined as a ‘diameter’ larger than 0.95 cm at the injection site [26,27].
(l. 97-100)
And the two first references are added:
Kinnett D. Botulinum toxin A injections in children: technique and dosing issues. Am J Phys Med Rehabil. 2004 Oct;83(10 Suppl):S59-64. doi: 10.1097/01.phm.0000141131.66648.e9.
And Sätilä H. Over 25 Years of Pediatric Botulinum Toxin Treatments: What Have We Learned from Injection Techniques, Doses, Dilutions, and Recovery of Repeated Injections? Toxins (Basel). 2020 Jul 6;12(7):440. doi: 10.3390/toxins12070440.
Brodsky MA, Swope DM, Grimes D. Diffusion of botulinum toxins. Tremor Other Hyperkinet Mov (N Y). 2012;2:tre-02-85-417-1. doi:10.7916/D88W3C1M
- When using r-FLACC, if the score was different from children and caregiver, which score was used?
Again, excellent point. Although parent- and self-reported pain, in general, is significantly correlated, parents tend to overestimate their child’s pain if self-reported pain is infrequent or mild and underestimate it if the self-reported pain is frequent or severe (Parkson et al (2013), Raiter et al. (2021)). Studies indicate, that there, as mentioned, is a general correlation between caregivers and children, but there can be a discrepancy of over- and underestimation of pain between child and caregiver depending on the severity of pain. Also, the estimation of pain for a caregiver would also often be an overall estimation of their child’s pain in presence of generalized neuropathic pain, additional pain from the gastrointestinal region and other causes. We were interested in the effect of localized treatment for localized pain thus we utilized an observation score, where we examined and evaluated the child ourselfs during passive range of motion and compared the ‘localized pain reaction’ by the ‘mod. r-FLACC’. Our evaluation of pain was as a consequence independent of the caregiver and relied on the behavioural signs in the facial expression, legs, activity, cry and consolability of the child, and one rater (the primary investigator) evaluated all the r-FLACC‘ responses from the child. If the primary investigator had doubts, then the r-FLACC videos were evaluated with the other investigators, thus independent of parental input.
We are aware that the gold standard of pain evaluation is the self-evaluation by the child, but for quite a number of our children, self-reporting of pain was not possible due to communication and/or cognitive impairments.
Thank you for this important comment, since we as a consequence will look into the potential differences in parental and self-reported pain in future studies. We have omitted the NRS and WB evaluations. We feel this is an important point. Thank you.
Raiter AM, Burkitt CC, Merbler A, Lykken L, Symons FJ. Caregiver-Reported Pain Management Practices for Individuals With Cerebral Palsy. Arch Rehabil Res Clin Transl. 2021 Jan 18;3(1):100105. doi: 10.1016/j.arrct.2021.100105
Parkinson KN, Dickinson HO, Arnaud C, Lyons A, Colver A; SPARCLE group. Pain in young people aged 13 to 17 years with cerebral palsy: cross-sectional, multicentre European study. Arch Dis Child. 2013 Jun;98(6):434-40. doi: 10.1136/archdischild-2012-303482

Reviewer 2 Report
ABSTRACT
Properly written, clearly summarize the paper. If possible, I'd shorten it a little bit to improve its readability.
INTRODUCTION
No remarks on this section. It provides clearly the study. background and summarizes in the last sentences the specific aim of the authors.
RESULTS
The Results section is clearly organized. The study methodology is clear and logical; the readability of the section is thus ensured but I would move the methods section before the results. The authors presents the outcome of the study in a very readable way ensuring the fluidity of the paper. Adequate use of tables to summarize the results.
DISCUSSION
The discussion section summarizes the main evidences emerging from the study and how they relate to other studies findings. A brief summary of the strengths and weaknesses of the paper is also provided.
Author Response
D
Thank you for an excellent summary and most relevant comment to the article ' Measuring effects on pain and quality of life after Abobotulinum Toxin A injections in children with cerebral palsy.'
We have revised the manuscript following your comments and will describe this in detail:
ABSTRACT
Properly written, clearly summarize the paper. If possible, I'd shorten it a little bit to improve its readability.
Excellent point. Thank you. The abstract is revised and shortened to 200 words as suggested to ensure and improve its readability.
INTRODUCTION
No remarks on this section. It provides clearly the study. background and summarizes in the last sentences the specific aim of the authors.
Thank you for this kind comment.
RESULTS
The Results section is clearly organized. The study methodology is clear and logical; the readability of the section is thus ensured but I would move the methods section before the results. The authors presents the outcome of the study in a very readable way ensuring the fluidity of the paper. Adequate use of tables to summarize the results.
Again, this is very kind of you.
DISCUSSION
The discussion section summarizes the main evidences emerging from the study and how they relate to other studies findings. A brief summary of the strengths and weaknesses of the paper is also provided.
Thank you for this kind comment.

Round 2
Reviewer 1 Report
The authors have revised the manuscript well according to the comments.